# *Withania somnifera* Extracts Promote Resilience against Age-Related and Stress-Induced Behavioral Phenotypes in *Drosophila melanogaster;* a Possible Role of Other Compounds besides Withanolides

**DOI:** 10.3390/nu14193923

**Published:** 2022-09-22

**Authors:** Helen Holvoet, Dani M. Long, Alexander Law, Christine McClure, Jaewoo Choi, Liping Yang, Luke Marney, Burkhard Poeck, Roland Strauss, Jan F. Stevens, Claudia S. Maier, Amala Soumyanath, Doris Kretzschmar

**Affiliations:** 1Institute for Developmental Biology and Neurobiology, Johannes Gutenberg-Universität Mainz, Hanns-Dieter-Hüsch-Weg 15, 55128 Mainz, Germany; 2Botanicals Enhancing Neurological and Functional Resilience in Aging, Botanical Dietary Supplements Research Center, Oregon Health and Science University, Portland, OR 97239, USA; 3Oregon Institute of Occupational Health Sciences, Oregon Health and Science University, Portland, OR 97239, USA; 4Department of Neurology, Oregon Health and Science University, Portland, OR 97239, USA; 5Linus Pauling Institute, Oregon State University, Corvallis, OR 97331, USA; 6Department of Chemistry, Oregon State University, Corvallis, OR 97331, USA; 7Department of Pharmaceutical Sciences, Oregon State University, Corvallis, OR 97331, USA

**Keywords:** ashwagandha, withanolides, sleep, depression-like state, locomotion, cognition

## Abstract

*Withania somnifera* (WS) extracts have been used in traditional medicine for millennia to promote healthy aging and wellbeing. WS is now also widely used in Western countries as a nutritional supplement to extend healthspan and increase resilience against age-related changes, including sleep deficits and depression. Although human trials have supported beneficial effects of WS, the study designs have varied widely. Plant material is intrinsically complex, and extracts vary widely with the origin of the plant material and the extraction method. Commercial supplements can contain various other ingredients, and the characteristics of the study population can also be varied. To perform maximally controlled experiments, we used plant extracts analyzed for their composition and stability. We then tested these extracts in an inbred *Drosophila* line to minimize effects of the genetic background in a controlled environment. We found that a water extract of WS (WSAq) was most potent in improving physical fitness, while an ethanol extract (WSE) improved sleep in aged flies. Both extracts provided resilience against stress-induced behavioral changes. WSE contained higher levels of withanolides, which have been proposed to be active ingredients, than WSAq. Therefore, withanolides may mediate the sleep improvement, whereas so-far-unknown ingredients enriched in WSAq likely mediate the effects on fitness and stress-related behavior.

## 1. Introduction

The average human lifespan has dramatically increased over the last decades worldwide, especially in industrial countries, due to improved nutrition and advances in medicine. According to the National Institute on Aging, almost 500 million people worldwide were 65 and older in 2006, and it is predicted that this number will double by 2030. Although we are living longer, this increased lifespan raises the risk to develop age-related ailments and diseases and to spend the additional years in poor health. Especially, the prevalence of chronic diseases, such as type 2 diabetes mellitus, cardiovascular disease, or arthritis, increases with age [1,2,3]. In addition, neurodegenerative diseases such as Parkinson’s disease and Alzheimer’s disease typically occur in the later years of life. A report in 2005 estimated that 24.3 million people had dementia worldwide, and this number increased to 46.8 million in 2015 [4]. Correlating with these numbers, the death rate from Alzheimer’s disease increased over the last decades, with Alzheimer’s disease now being the sixth leading cause of death in the United States [5]. In addition to increasing the risk for developing these diseases, aging is also associated with a decline in locomotion, sleep disruptions [6,7,8], changes in cognition [9,10], and increasing anxiety and depression [11,12]. Although these are not pathogenic conditions, they nevertheless impair the well-being of the elderly and often cause unwanted changes in their lifestyle. These conditions are often treated pharmacologically; however, the outcomes are mixed, and many drugs have side effects. For example, sleep disruptions can be treated with medications such as trazodone, benzodiazepines, and nonbenzodiazepine hypnotics [6], but they also result in dizziness, hypotension, and an increased risk of falls [6,7]. Similarly, cholinesterase inhibitors to treat cognitive decline and dementia also have notable side effects, including headaches, dizziness, and insomnia [13,14,15]. Although these medications provide a treatment for severe or pathogenic cases, they are not an option for promoting resilience and healthy aging.

Therefore, there has been a growing interest in nutritional supplements to promote resilience and healthspan. In 2020, the global market for nutritional supplements was valued at USD 310 billion, and this market segment is predicted to grow by 6.2% annually. In addition to vitamins and minerals, these supplements also include herbs that have been used in traditional medicine, such as Ayurveda, for centuries [16,17,18]. One of the herbs that is used in Ayurvedic and other traditional medicines to promote health and longevity is *Withania somnifera* [16,18]. *Withania somnifera* (L.) Dunal is a small shrub in the nightshade (Solanacea) family, and is also known as ashwagandha, winter cherry, or Indian ginseng. In particular, the dried root has been used for the treatment of arthritis, anxiety, and sleep disorders [16,19,20]. *W. somnifera* (WS) has now also been studied in several placebo-controlled human trials. Meta-analyses of these studies showed that WS supplementation can indeed promote muscle strength and physical performance, reduce anxiety and stress levels, and improve sleep, while no serious side-effects were reported in any of these studies [21,22,23,24]. Although this supports beneficial effects, the individual trials varied in the selected study population (for example age, gender, or including patient groups), preparation of the WS extracts, and phytochemical composition of the plant material, complicating the evaluation of the data [25]. Commercially available WS preparations are even less standardized, varying widely in the amount of WS they contain, percentage of known compounds, how they are prepared, and what other ingredients are included.

In this study, we therefore used the *Drosophila melanogaster* model to test effects of analytically controlled WS preparations. *Drosophila* has been used in research for over a century, and this model allows for the control of the genetic background as well as the environment [26,27,28]. Various behavioral assays have been established in *Drosophila*, and, like humans, flies show age-related changes in locomotion, cognition, and sleep patterns [29,30]. We have therefore used *Drosophila* to determine effects of our WS extracts on locomotion, sleep, and anxiety and stress levels in a controlled experimental system.

## 2. Materials and Methods

### 2.1. Fly Stocks

All experiments were performed with *Drosophila melanogaster* wild-type Canton-Special strain (CS) obtained from Martin Heisenberg (University Würzburg). Flies were maintained on standard *Drosophila* food at 25 °C under a 12:12 h light–dark cycle. Food was prepared by mixing the WS extracts, provided as 10× aqueous stock solutions, into warm (liquid) standard *Drosophila* food. Control vials were prepared by mixing the same amount of water into the food.

### 2.2. Plant Material

*W. somnifera* plants were grown at Oregon’s Wild Harvest (OWH), Redmond, Oregon, USA. The roots were harvested (OWH lot number 201000162) and obtained in bulk, and voucher samples were deposited in the Oregon State University (OSU) Herbarium (voucher number OSC-V-265405) and in our laboratories (Code number BEN-WS-8) at Oregon Health and Science University (OHSU). Aqueous extracts of the dried ground roots were prepared by boiling in deionized water under reflux for 90 min at a ratio of 160 g material: 2 L water. The mixtures were then filtered through a kitchen sieve while still warm to remove larger plant particles. The extracts were centrifuged at 3750 rpm for 10 min on a benchtop centrifuge (Beckman GS-6R); the pellet, which contained finer plant particles, was discarded and the supernatant filtered through Whatman filter paper (Grade 1, 90 mm). This extract was then frozen and lyophilized into a powder on a Virtis lyophilizer (Phase 1, 115 V, 20 amps). The yield of dried water extract was about 10.4% by weight of the original plant material. A 70% ethanol extract was prepared by boiling the plant material under reflux with 70% ethanol in water for 90 min. The extracts were filtered through a sieve and centrifuged, and the supernatant was filtered through filter paper as described for the aqueous plant extracts above. Ethanol was then removed from the filtrate under vacuum on a rotary evaporator, and the remaining predominantly aqueous solution was frozen and lyophilized as described for the aqueous extract. The yield of dried 70% ethanol extract was about 9.5% by weight of the original plant material. Each extraction was given a specific lot number and stored at −20 °C until use. The specific extracts used in these studies were WSAq-2 and WSAq-9 (aqueous extracts of BEN-WS-8 root) and WSE-2 and WSE-5 (70% ethanol extracts of BEN-WS-8 root). Voucher samples of the extracts are stored in our laboratories under these lot numbers. Chemical fingerprints of the extracts were acquired by LC-HRMS (Appendix A).

### 2.3. Analysis of the WS Extracts and Drosophila Food by LC-MRM-MS

#### 2.3.1. Analysis of the WS Extracts (WSAq and WSE)

Liquid chromatography coupled to multiple reaction monitoring mass spectrometry (LC-MRM-MS) analysis of selected WS phytochemical markers in extracts was performed as described by Choi et al. (70th ASMS conference on Mass Spectrometry and Allied Topics, Minneapolis, MN, USA, 5–9 June 2022, ThP 399). To quantify withanolides markers in aqueous or 70% ethanol extract, 0.5 mg-each extracts powder was reconstituted in 1 mL 70% methanol with 0.1% formic acid, with the addition of digoxin-d_3_ (1 µg/mL) as an internal standard. The resuspended samples were sonicated for 30 min at room temperature and centrifuged (14,000× *g* for 10 min), and supernatants were analyzed by LC-MRM-MS (Appendix A).

#### 2.3.2. Analysis of the *Drosophila* Food

Analysis of selected WS phytochemical markers in fly food was performed using the same LC-MRM-MS method as described for the analysis of the extracts (see Section 2.3.1). Vials with control and supplemented food were frozen immediately after preparation or after 7 days at 25 °C in the fly room. *Drosophila* food was prepared for phytochemical analysis by weighing 50 mg of the sample and dissolving in 1 mL 70% methanol with 0.1% formic acid, with the addition of digoxin-d_3_ (1 µg/mL) as the internal standard. Samples were then vortexed for 30 s and sonicated for 15 min twice. Samples were centrifuged, and supernatant was used for LC-MRM-MS analysis.

### 2.4. Measuring Food Intake

To determine whether flies showed any feeding preferences to the various concentrations of WS extracts, we adapted a protocol from the literature [31]. Age-matched flies were transferred to food with three formulations: regular food with neither dye nor WS extracts, food with the addition of just dye (FD and C Blue #1 (1%, *w*/*v*)), and food with the addition of both dye and WS extract at different doses. After being allowed to feed for 1 h, flies were separated by sex and frozen at −80 °C. Three independent feeding experiments were performed for each condition. To determine the amount of dye present in the flies, we measured absorbance at 630 nm (A630) using the UV-vis option on a NanoDrop spectrophotometer. Five flies from each experiment were homogenized in 50 μL phosphate-buffered saline (PBS) plus 1% Triton X-100m then centrifuged to clear debris. An aliquot of supernatant was taken, centrifuged again, and finally measured for A630 values. A standard curve was made by mixing a known percentage of dye with PBS, 1% Triton X-100, then measuring A630 values for six concentrations. A630 values from the fly samples were subtracted by the fly samples containing no dye, then plugged into the slope equation generated by the standard curve.

### 2.5. Phototaxis Assay

Newly eclosed flies were collected daily and transferred to control food or food supplemented with the WS extracts, and fresh vials were provided every 7 days. When using aged flies, the flies were kept on standard food for four weeks before the two week treatment. Males and females were aged together but tested and analyzed separately. Fast phototaxis assays were conducted in the dark as previously described in Dutta et al. [32] and Bolkan et al. [33] using the countercurrent apparatus described by Benzer [34] and a single light source. A detailed description of the experimental conditions can be found in Strauss and Heisenberg [35]. Briefly, flies were transferred to the apparatus in groups of 10–15 flies, shaken to the bottom of the vial and allowed to transition toward the light in 5 consecutive runs, each lasting 6 s. Flies were then scored based on the tube they were contained in at the end of the final run. Statistical analyses were performed using GraphPad (v.5 for windows, San Diego CA, USA). Normal distribution was addressed with the D’Agostino and Pearson omnibus test, and due to the nonparametric distribution, Mann–Whitney tests were used to determine significance between two samples, and Kruskal–Wallis ANOVA (with built-in Bonferroni posthoc) was used when comparing multiple groups. Significance levels are indicated by asterisks, with * *p* < 0.05, ** *p* < 0.01, *** *p* < 0.001.

### 2.6. Sleep Assay

As with the phototaxis assays, newly eclosed flies were collected daily, aged on standard food, and transferred to control food or food supplemented with the extracts for two weeks before being tested. Males and females were aged together before assessing sleep using the *Drosophila* Activity Monitoring Systems (DAMS) as described in Cassar et al. [36] and Metaxakis et al. [37]. Flies were placed individually in glass tubes with standard *Drosophila* food placed in one end and the other end sealed with a short piece of yarn (approx. 1.5 cm). In the experiment with continuous feeding, the flies were fed either standard control food or food supplemented with WS extract during the sleep studies. Glass tubes were placed in DAMS model DAM2 (Trikinetics, Waltham, MA, United States). Locomotor activity of the flies was recorded once every minute for 8 days in 12 h light/12 h dark cycles. Due to the flies having to adapt to the new environment, data from the first day were not included in the analyses. A sleep bout was regarded as a period of 5 min or more with no movement detected. Data were analyzed using ClockLab (v.6.1.02 for Windows, Actimetrics, Wilmette, IL, USA), and a 2-tailed unpaired t-test with Welch’s correction was performed to compare sleep activity between two experimental groups, and a one-way ANOVA with Dunnett’s post-tests were used to compare multiple groups to a control. Significance levels are indicated by asterisks, with * *p* < 0.05, ** *p* < 0.01, *** *p* < 0.001.

### 2.7. Stress Protocol

Flies were collected and their wings shortened when 2 to 3 days old to prevent flying during the gap climbing and stop-for-sweet assay (see Section 2.8 and Section 2.9). Cohorts of 10–20 flies were then aged on fly vials for a total of 10 days with or without the root extracts, and the vials were replaced with fresh ones at day 5 and day 10. Stress was then applied with repetitive phases of 300 Hz vibrations with cohorts of 10–20 flies confined to empty, narrow tubes during daytime (usually 8 am to 6 pm), as described earlier [38]. After the stress application, flies were transferred back to standard food or WS-supplemented food for the night. For the non-vibrated control, flies were confined to the same empty, narrow plastic tubes and placed next to the vibrating device for the same amount of time. For analyzing prophylactic capacity of WS extracts against stress, flies were kept on standard food during the three days of stress application. For continuous treatment, flies were returned to food supplement with WS in each rest period of the stress protocol.

### 2.8. Gap Climbing Assays

After ten days on WS supplemented food or standard food, flies were tested for their motivation to initiate a climbing attempt at a 4.5-millimeter-wide gap. Each fly was allowed to perform ten approaches to the gap. Only flies that showed four or more climbing attempts in this pretest (PT) were included in the stress protocol. The post-stress tests (T1) were performed on day 13 of the experiment after a short resting period on standard food. An attempt to climb the gap was defined by the stereotypical leg-over-head behavior [38]. Comparative statistical analyses were carried out using RStudio. Shapiro–Wilk’s tests were used to test for normal distribution, and a pairwise t-test with Bonferroni–Holm correction was applied for multiple comparisons where appropriate. For nonparametric data, a pairwise Wilcoxon test was applied with built-in Bonferroni–Holm correction to compare three or more experimental groups/conditions against each other. Significance levels are indicated by asterisks, with * *p* < 0.05, ** *p* < 0.01, *** *p* < 0.001.

### 2.9. Stop-For-Sweet (S4S) Assay

Flies with their wings cut were subjected to the same feeding and stress protocol as outlined above. However, on the last day of stress application, flies were placed in empty vials to keep them food-deprived for at least 8 h. The S4S paradigm was performed as described in Ries et al. [38]. Individual flies were confined to rectangular chambers of 55 × 20 mm^2^ that were cut out of a 3-millimeter-thick white foam board with a clear plastic bottom. The chambers were covered with a cut-to-size filter paper on which a 5-millimeter-wide trace of glycerol (99.5%) had been applied along the midline of the paper with a fine paintbrush. To induce negative geotaxis, the flies were shaken to the bottom of the chamber, and the chamber was turned (110°–120°) to make the flies walk at a 90° angle upwards on the filter paper. Each fly was observed to determine whether it kept walking up the filter paper or stopped and extended its proboscis to the glycerol. After proboscis extension (or continuous walking) was noticed by observation, the fly was immediately shaken down to prevent ingestion, and the protocol was repeated ten times for each fly. Statistical analyses were carried out using RStudio. Shapiro–Wilk’s tests were used to test for normal distribution, and a pairwise t-test with Bonferroni–Holm correction was applied for multiple comparisons where appropriate. For nonparametric data, a pairwise Wilcoxon test was applied with built-in Bonferroni–Holm correction to compare three or more experimental groups/conditions against each other. Significance levels are indicated by asterisks, with * *p* < 0.05, ** *p* < 0.01, *** *p* < 0.001.

## 3. Results

### 3.1. WS Is Stable in Standard Fly Food 

To determine whether WS compounds were present and stable over the 7-day period that a fly food vial was used, we measured phytochemical markers in the food samples by LC-MRM-MS. The markers were present in the food after preparation, and six of the seven phytochemical markers were stable for 7 days in *Drosophila* food. A slight increase in concentration (3 to 20%) due to moisture evaporation over the seven-day periods in which the food was in the open and at room temperature was observed (Figure 1). Only Withaferin A showed less-consistent behavior in the food matrix, possibly due to its known chemical reactivity [39]. 

Comparing the phytochemical markers in the fly food supplemented with 0.5 mg/mL WSAq-9 and WSE-2 also revealed that the levels of withanolides in the 70% ethanol extract are higher than in the water extracts, approximately increasing between three- and six-fold (values of the measurements are shown in Appendix A). The increase in withanolides in the fly food supplemented with the ethanol extract is comparable to what we observed when measuring the withanolide levels in the powdered extract (Appendix A). 

### 3.2. WS Supplementation Does Not Lead to a Decrease in Consumption

Next, we tested whether the flies consume the food containing the different WS extracts to exclude that a failure of the supplemented food to promote resilience is due to avoiding eating the food. Measuring the food intake using a colorimetric assay, we found that neither the addition of WSAq-9 nor WSE-2 reduced the consumption (Figure 2). While there was no significant change in males, females showed an increase in consumption of food containing 0.5 mg/g of WSE-2.

### 3.3. WS Extract Promotes Performance in Fast Phototaxis Assays in Females

As mentioned above, *Drosophila* shows a decline in locomotion and cognition with age [29,30]. To determine whether WS provides resilience against these age-related changes, we used the fast phototaxis test, in which the flies have to locate and run toward a light source after being startled. We previously showed that supplementing standard food with 0.5 mg/g or 2.0 mg/g of the WS water extract improved the performance of the *Drosophila sniffer* mutant that shows increased oxidative damage [40] in this assay [41]. We therefore also initially used these concentrations to assess whether this had positive effects on the age-related decline in performance. In addition, we included a concentration of 5.0 mg/g because our previous experiments with *sniffer* suggested that a higher dose may be more protective. As shown in Figure 3A,B, male and female flies show a continuous decline in performance in the fast phototaxis assays with age. Although our aim was to identify treatments that could improve healthy aging, we first tested whether we could detect effects (including deleterious effects) in young flies, but as shown in Appendix A, there was no difference between treated and untreated 14-day-old flies. We then decided to focus on six-week-old flies that showed a significant decline in performance in this assay. In addition, we kept the flies on standard food during development and the first four weeks of adulthood because we aimed to identify protective effects when given during the later phases of life. After aging for four weeks on normal food, the flies were transferred to the supplemented food for two weeks before being tested at six weeks of age. While we did not detect an improvement in males given WSAq-9 (Figure 3C), we did detect protective effects in females with either 0.5 mg/g or 2.0 mg/g WSAq-9 (Figure 3D). To test whether a higher dose may have a better effect, we also tested 5.0 mg/g WSAq-9, but again, the behavior of males was not improved, and although treated female flies did perform better than controls, this did not reach significance. 

Using the 70% ethanol extract (WSE), we again did not detect effects in male flies at any of the concentrations used, but females performed significantly better when given 2.0 mg/g (Figure 3E,F). As shown in Figure 1 and Appendix A, the levels of withanolides are about 3–6 times higher in WSE-2 compared to WSAq-9. To exclude that WSE-2 was less protective due to a negative effect of high levels of withanolides, we included a lower dose of 0.05 mg/g WSE-2. However, this concentration also had no effect, arguing against a toxic effect of increased levels of withanolides. Together, this suggests that a compound or compounds that are increased in the aqueous extract compared to the ethanol extract mediate the protective effect.

### 3.4. WSE-2 Improves Sleep, Whereas WSAq-9 Does not

Another age-related phenotype conserved between *Drosophila* and humans is poor sleep. As in humans, this is detectable as increased sleep fragmentation with more sleep periods which are, however, shorter [42]. WS has traditionally been used to treat sleep disorders [16,19], and although only a few clinical trials have been performed, they do support a beneficial effect of WS on sleep quality [25]. To address this issue in our model, we first confirmed that we can detect age-related sleep fragmentation by determining the number of daily sleep bouts and their lengths. As shown in Figure 4, we found that 6-week-old male flies show a significant increase in sleep fragmentation (Figure 4A,B), and females also showed a tendency toward an increased number of sleep bouts (Figure 4D). Both males and females showed a significant decrease in nighttime sleep (Figure 4C,F). 

Therefore, we again used 6-week-old flies to determine the effects of WS, keeping the flies for 4 weeks on standard food followed by 2 weeks on supplemented food before performing the sleep studies. During the 8 days measuring sleep, the flies were kept on standard food. Using WSAq-9, we found a reduction in sleep bout number in males but only at the concentration of 2.0 mg/g, and this was not accompanied by a significant increase in sleep bout length (Figure 5A,B). Measuring the length of daytime, nighttime, and total sleep time during a day also revealed no significant differences, with the exception of males treated with 5.0 mg/g WSAq-9, which actually showed less nighttime sleep (Figure 5C). Analyzing sleep in females, we did not detect any significant changes at any of the concentrations used (Figure 5D–F).

Repeating this experiment with WSE-2, we did observe a reduction in the number of sleep bouts but an increase in their length in males. This appeared to be dose-dependent and did reach significance at the highest dose of 2.0 mg/g (Figure 6A,B). This was accompanied by longer daily total sleep time due to an increase in nighttime sleep at 0.05 mg/g and 2.0 mg/g (Figure 6C). In contrast to males, WSE-2 supplementation did not improve sleep fragmentation in females (although it did increase sleep bout length at 0.05 mg/g, Figure 6D,E), but it also prolonged nighttime sleep at the concentrations of 0.05 mg/g and 2.0 mg/g (Figure 6F). To address whether continuing supplementation during the time the flies are in the sleep monitors would improve the effects, we provided food containing 0.5 mg/g and 2 mg/mL WSE-2 during sleep monitoring. As shown in Appendix A, we obtained similar results as in the previous experiments for males with less sleep fragmentation and increased sleep time that was now also significant at 0.5 mg/g. However, when treating females, it did result in a more fragmented sleep pattern with more sleep bouts that were shorter and less sleep time. Together, this shows that WSE-2 can improve sleep in aged flies but only in males, whereas WSAq- 9 had no effect on either males or females.

### 3.5. WSAq-2 and WSE-5 Promote Resilience to a Stress-Induced Depressive-like State

As mentioned above, WS has been used to reduce anxiety and stress levels [21,22,23], and we therefore also tested our plant preparations for effects on a depression-like state (DLS) in *Drosophila*. Stress was induced in 12–13-day-old flies by phases of vibrations of about 10 h for three consecutive days, after which the flies were tested for their motivation to climb an insurmountable gap as described previously [38]. Because we did detect beneficial effects of WSAq-2 at a concentration of 0.5 mg/g in the phototaxis assays in males, we used this concentration to treat males either before applying the stress (prophylactic treatment, pro.) or continuously (con.) during the entire experiment (Figure 7A). To determine whether treatment with WSAq-2 affected their willingness to initiate climbing attempts, we first tested the flies before stress application (pre-test; PT). As shown in Figure 7B, the prophylactically treated male flies did not behave differently than the controls in this pre-test, and the continuously treated flies initiated slightly more climbing attempts. Control males performed significantly worse after the stress was applied (T1), with 30% fewer climbing attempts compared to the pre-test. The flies receiving WSAq-2-supplemented food, either prophylactically before or continuously during the stress paradigm, did not show any decline in their performance and were not significantly different from the performance in the pre-test (Figure 7B), indicating that WSAq-2 conveys resilience to chronic stress. To support this idea, we also tested treated and control flies for anhedonia in the so-called stop-for-sweet (S4S) paradigm, where stressed flies performing negative geotaxis ignore the sweet stripe in their path. Similar to the results in the climbing assay, both treatment strategies using WSAq-2 prevented the stress-induced reductions in stopping for the sweet taste (Figure 7C).

Next, we tested whether WSAq-2 might also increase the resilience of females against stress. As in males, both the prophylactic as well as the continuous treatment increased the climbing attempts after stress compared to the stressed control group (Figure 7D). Although prophylactic treatment with WSAq-2 did improve the motivation to climb, these females still showed significantly fewer climbing attempts than before being stressed, whereas the females with continuous supplementation performed as well as in the pre-test. When testing in the stop-for-sweet assay, prophylactic treatment had no effect, but continuous treatment did increase the number of stops in stressed females (Figure 7E). These results indicate a slight sexual dimorphism in WSAq-2-treated flies and highlight the benefits of continuous food supplementation. 

To investigate whether the WS extraction method affected the efficacy, flies were also treated with 0.5 mg/g of WSE-5 using the same treatment paradigm (Figure 8A). With this extract, we found that WSE-5-treated males had a significantly higher initial climbing motivation during the pre-vibration test (PT) compared to the control group. We also found that it protected the males from the reduction in climbing following stress when given prophylactically or continuously, and in both cases, the flies performed as well as in the pre-test (Figure 8B). However, in the stop-for-sweet test, only the continuous feeding had a protective effect (Figure 8C). In females, prophylactic treatment could not improve the stress-induced reduction in climbing but did increase the stops at the sweet-tasting stripe (Figure 8D,E). In contrast, with continuous supplementation, the female flies performed significantly better in both tests. These results again indicate a sexual dimorphism in favor of male flies. They also show that continuous supplementation is more efficient than prophylactic treatment alone, which was to be expected, and they further highlight the importance that the extraction method has an effect on biological activity. 

Due to our finding in the phototaxis experiments that increasing the dose of the 70% ethanol extract to 2.0 mg/g did result in a significant improvement in females, we also tested this dose in the stress-related behavior assays. As shown in Figure 9B, it still improved the performance in the gap-climbing test in males, confirming that the higher dose had no toxic effect. It now also resulted in a significant improvement in the stop-for-sweets test when given prophylactically (Figure 9C). Similarly, in females, prophylactic treatment with 2.0 mg/g WSE-5 provided resilience in the climbing assay with no significant difference between the pre- and post-test (Figure 9D). As with the 0.5 mg/g, the 2.0 mg/g dose of WSE-5 restored the performance in the stop-for-sweets test (Figure 9E). 

## 4. Discussion

WS has traditionally been used for centuries mostly by dissolving powdered roots or leaves in water or milk [43]. Its efficacy has also been shown in clinical trials, mostly using aqueous or alcoholic extracts, by improving cognition and sleep quality or reducing stress and anxiety [22,24,25,43]. However, the study populations have included a wide range of ages, often relied on subjective measurements, and to our knowledge, none have compared different extraction methods. We have therefore used the *Drosophila* model, which allows us to tightly control for age, environment, and genetic background. Our results show that supplementation of the food with WS extracts can provide resilience to age-related deficits in *Drosophila*. However, the effects depend on the preparation of the extracts and can be gender specific. This was the case when performing fast phototaxis assays, which allow one to test for locomotion fitness and cognitive function by orienting and walking toward a light source. As shown in Figure 1A,B, both males and females show an age-related decline in these tests, but WS treatment only improved the performance in females. Furthermore, the efficacy was different when comparing the aqueous (WSAq) extract to the 70% ethanol (WSE) preparation of WS. While WSAq already had an effect at 0.5 mg/g, WSE was only effective at the higher dose of 2.0 mg/g. To our knowledge, the effects of WS on locomotion during normal aging has not been studied so far, but locomotion tests have been performed in mutants showing locomotion phenotypes. Using a negative geotaxis test, which measures the time the flies need to run up a certain distance in a horizontal tube, has shown that WS improves the performance in a *Drosophila* model of amyotrophic lateral sclerosis (ALS) and Parkinson’s disease [44,45,46]. These studies have used methanolic root extracts at 1 mg/g and 10 mg/g, respectively (in [44]), and only males were included. Because we did not observe effects in males, this suggests that WS may improve fitness in males only in the context of a disease but not during normal aging. 

Assessing the effects of WS on another age-related phenotype, sleep disruptions, we again observed different effects of the preparation and gender of the flies. In these experiments, WSE provided resilience against age-related sleep fragmentation in males by reducing the number of sleep bouts but increasing their length. It also increased the nighttime sleep. In contrast, WSAq neither improved sleep fragmentation nor did it increase the time spent asleep. As shown in Figure 4, females did not show a significant increase in sleep fragmentation when aged, and therefore, it might not be surprising that neither WSAq nor WSE had an effect. However, 42-day-old females do exhibit a decrease in nighttime sleep (Figure 4), and this was improved by WSE. We also tested whether continuous supplementation with WSE during the time the flies spent in the sleep monitors (in addition to the two weeks before the monitoring) provided better protection against sleep disruption. While this was the case for males, which now showed a significant improvement of sleep fragmentation also at the lower dose of 0.5 mg/g, it actually increased sleep fragmentation in females and reduced sleep time. Although it is possible that the increased locomotion in females after WSE supplementation negatively affected sleep, this appears unlikely because WSE only increased locomotion at the higher dose of 2 mg/g. In addition, WSAq had stronger effects on locomotion but did not affect sleep. A sleep improvement in males has also been described in the ALS model, where a 1 mg/g methanolic extract (unknown if root or leaves were used) increased nighttime sleep ([46], females were not included). A study on the effects of WS in sleep-deprived flies only tested females, and in this case, an ethanolic root extract (1 mg/g) increased total sleep time and reduced latency [47]. This publication also included effects on sleep in an Alzheimer’s disease (AD) model, but WS supplementation had no beneficial effect on sleep in this model. 

The last age-related behavior we included was depression, which often affects the elderly, and which also has been described to be improved by WS [25,43]. To model this in flies, we induced a depression-like state, which is defined by a lack of motivation to perform voluntary behaviors after chronic uncontrollable stress. Determining the effects on climbing attempts, which were reduced after the stress, we found that in males, the aqueous extract completely restored the behavior to the levels before the stress was applied. This was the case when WSAq was either only given prophylactically before the stress paradigm or continuously. The same effect was detectable on the stop-for-sweet behavior, a measurement for anhedonia in flies. While WSAq did improve the climbing behavior in females too, it only partially restored it with the prophylactic treatment. The stop-for-sweet behavior was not improved by the prophylactic treatment in females. Treatment with WSE again restored the climbing behavior in males but only had an effect in females when given continuously or at the higher dose of 2.0 mg/g. Together, this suggests that males benefit more than females, and although prophylactic treatment can provide resilience against stress, continuous supplementation during stress is more effective. Furthermore, the aqueous WS extract was more effective in reducing stress-induced behavioral changes than WSE. This is similar to the results in the fast phototaxis tests where WSAq was protective at a lower dose than WSE. This was somewhat unexpected, because our analyses of the supplemented food showed that WSE had approximately three- to six-fold higher levels of the measured withanolides compounds. The withanolides have been proposed to be the major pharmacologically active compound in WS, although only a few studies have specifically addressed their biological activity. Withanone and withanoside IV have been shown to improve Aß-induced phenotypes [48,49], and Withaferin A has been connected to cell viability and proliferation in connection to cancer [50,51,52,53]. Our experiment suggests that other compounds that are enriched in the aqueous extracts are more beneficial in improving stress-induced behavioral changes and locomotion/fitness in aged flies. This is also supported by experiments in mice in which a withanolide-free extract decreased stress-induced changes in body weight, core temperature, and behavior [54]. In addition, a root extract improved behavior changes after learned helplessness in rats [55,56].

Although “*somnifera*” means sleep-inducing, and WS has been used for this purpose in traditional medicine [43], not many experimental studies have addressed this property. Like the study in *Drosophila* mentioned above [47], studies in mice and rats found improved sleep in the context of sleep deprivation but have not addressed age-related sleep disruptions [20,57]. In randomized human trials, WS has mostly been shown to be beneficial in insomnia patients, though some have reported sleep improvements in healthy subjects [22]. However, sleep disruptions and anxiety/stress are bidirectionally connected [58,59], making it difficult to determine whether the improvements in these studies were due to effects on sleep or on anxiety/stress. To our knowledge only one placebo-controlled study has investigated the effects of WS supplementation on sleep in the elderly and described a better sleep quality by questionnaire [60]. This study used capsules from root extracts treated with milk, and the extraction method is therefore not comparable to ours. In our model, the 70% ethanol-extracted preparation reduced the sleep-fragmentation in aged males and increased nighttime sleep in both males and female, while the water extract did not. Due to the higher content of withanolides in the 70% ethanol extract, the sleep-promoting effects of WS could therefore be mediated by a group or by an individual withanolide. However, it appears unlikely that the withanolides mediate the effects on fitness and cognition due to the ethanol extract being much less protective in the fast phototaxis assay than the water extract. This suggests that other compounds present or enriched in the water extract are major factors in providing resilience in this assay. The water extract was also more effective in reducing stress-induced behavioral deficits, although the ethanol extract was protective at the higher dose. This indicates that compounds present at higher levels in the water extract may be the major protective factors, although ethanol soluble factors such as the withanolides could contribute to the protective function. Therefore, our studies suggest that different compounds in WS may be effective on different behaviors. This provides the need to perform future studies to identify especially the compounds that are enriched in the water extract and to determine the compounds that are protective against specific behavioral deficits. Furthermore, even for the best-studied compounds, the withanolides, the mechanistic functions are not well-studied [61]. Withanolides and specifically Withaferin A have been shown to activate anti-inflammatory pathways and exhibit anticancer functions by inhibiting mitosis and cytoskeletal changes, that would enable growth and malignancy [62,63]. WS extracts have also been shown to have antioxidant functions, and their effects on anxiety and depression have been connected to interactions with Gamma-Aminobutyric Acid (GABA)-related activity [25,64,65]. Effects on GABA might also underlie the sleep-promoting function of WS [20], but it has also been described that WS can restore age-related changes in circadian clock genes [66]. Future studies are required to determine whether changes in these pathways play a role in the protective effects of WS in our assays and which are the compounds mediating these effects.

## 5. Conclusions

We show that WS can provide resilience against age-related decline in fitness and sleep disruptions in the *Drosophila* model. It also protects against stress-induced behavioral changes. The efficacy is dependent on gender and the extraction method. The aqueous extract is more effective in promoting fitness and reducing stress-induced behavioral changes than the ethanol extract, although it has four to six times lower levels of withanolides. This suggests that other compounds in WS mediate these effects. In contrast, the ethanol extract improved sleep, indicating that withanolides might promote sleep.

## Figures and Tables

**Figure 1 nutrients-14-03923-f001:**
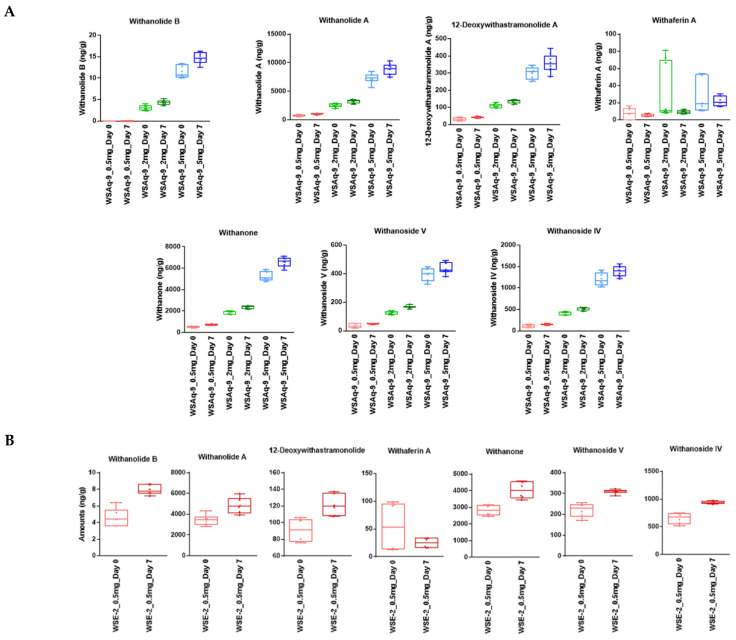
Stability of phytochemical markers in *Drosophila* food kept for seven days at 25 °C. (**A**) WS phytochemical marker levels (ng/g of food) measured for day 0 (light color) after 7 days (dark color) in food containing 0.5 (red), 2 (green), or 5 (blue) mg WSAq-9 extract/g of food. (**B**) WS phytochemical marker levels (ng/g food) measured for day 0 (light) and after 7 days (dark) in food containing 0.5 mg WSE-2 extract/g food. For WSAq-9, n = 3 with triplicate runs. For WSE-2, n = 2 with triplicate runs. The horizontal bars in the box plots represent the medians, boxes the 25% and 75% quartiles, and whiskers the max. and min. values.

**Figure 2 nutrients-14-03923-f002:**
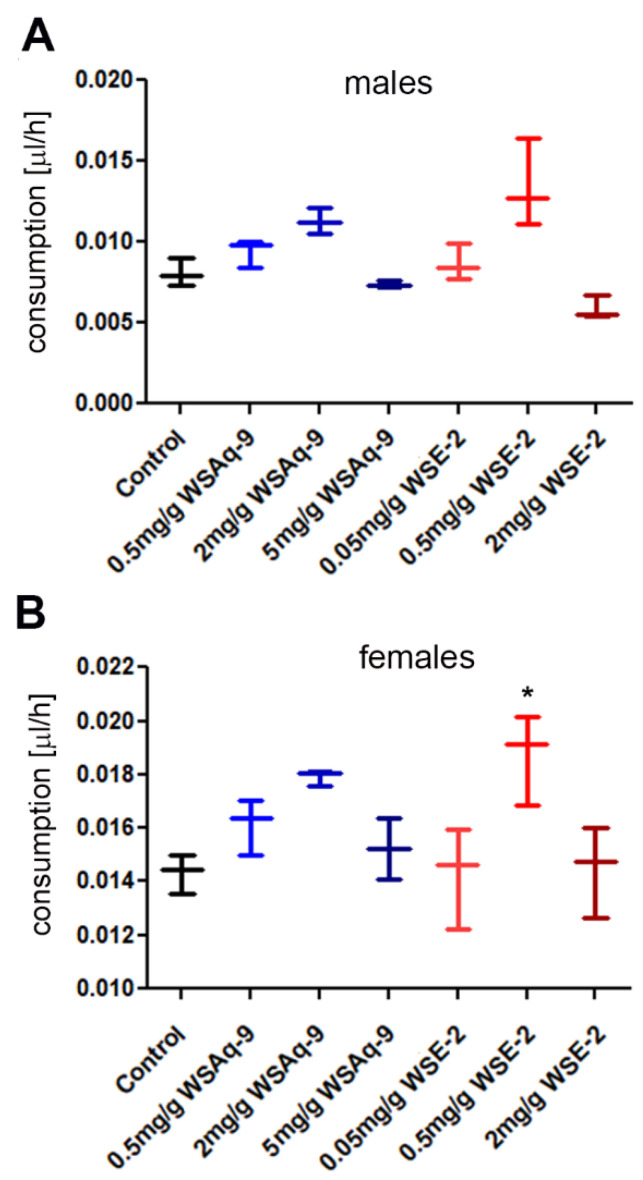
Food intake is not reduced by supplementation with WSAq-9 or WSE-2. (**A**) Males show no significant change in intake at any concentration of WSAq-9 or WSE-2. (**B**) Females also showed no change in intake of the supplemented food, with the exception of WSE-2 at 0.5 mg/g which was consumed slightly more. A D’Agostino and Pearson omnibus test showed that the data were not normally distributed, and the significance was then determined by Mann–Whitney tests comparing supplemented food to the control. Three independent measurements with 5 flies each were performed. The horizontal bars in the box plots represent the medians, boxes the 25% and 75% quartiles, and whiskers the max. and min. values. * *p* < 0.05.

**Figure 3 nutrients-14-03923-f003:**
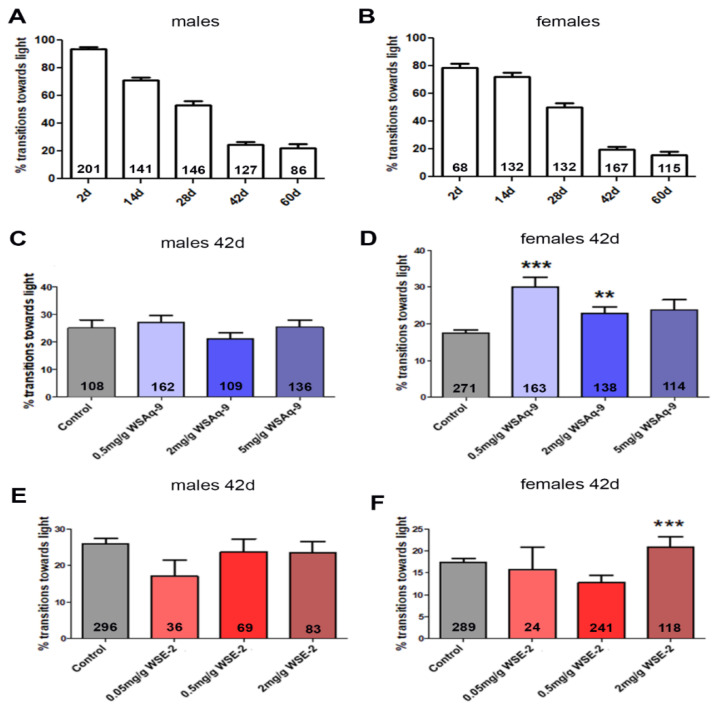
Performance in the phototaxis assay declines in males (**A**) and females (**B**) with age. (**C**) WSAq-9 does not improve the performance in males at any concentration tested. (**D**) Performance is improved in females fed with food supplemented with 0.5 mg/g and 2.0 mg/g WSAq-9. (**E**) WSE-2 does not increase the phototaxis response in males at any of the concentrations tested. (**F**) Females perform better when given 2.0 mg/g WSE-2. Kruskal–Wallis ANOVA with built-in Bonferroni post-hoc test was used for statistical analyses. Bars represent means and error bars SEMs. The number of tested flies is given in the bars. ** *p* < 0.01, *** *p* < 0.001 compared to controls on regular food.

**Figure 4 nutrients-14-03923-f004:**
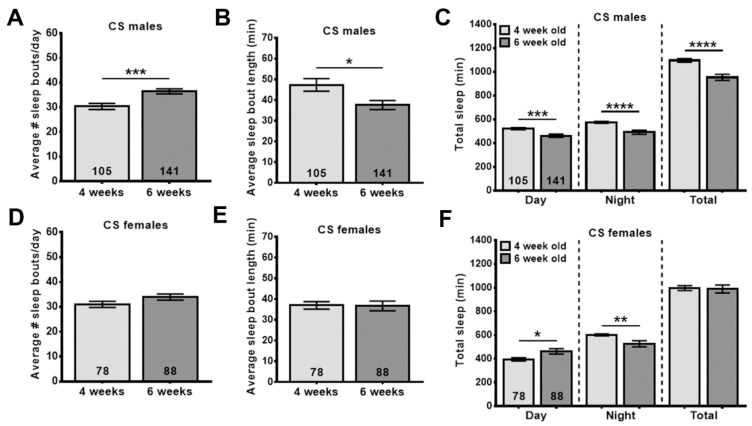
Aged flies show sleep disruptions. Additionally, 42-day-old males show a significant increase in sleep bout number (**A**) while sleep bout length is decreased (**B**) when compared to younger flies (28 d). (**C**) Both daytime and nighttime sleep is reduced in the older flies. 42-day-old females do not show significant changes in their sleep pattern, though the sleep bout number is slightly increased (**D**,**E**). However, 42-day-old female flies show decreased nighttime sleep compared to 28-day-old flies, whereas their daytime sleep is increased (**F**). Sleep was analyzed starting at the indicated age. The number of analyzed flies is given in the bars, and a Student’s t-test was used to determine significance. Bars represent means and error bars SEMs. * *p* < 0.05, ** *p* < 0.01, *** *p* < 0.001, **** *p* < 0.0001.

**Figure 5 nutrients-14-03923-f005:**
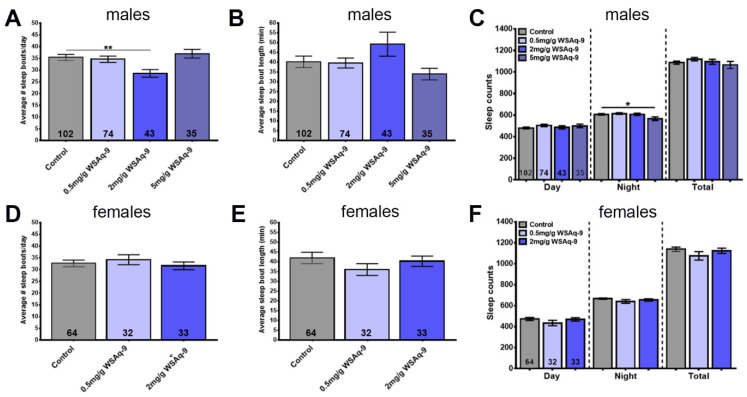
WSAq-9 does not improve sleep. Males show a reduction in sleep bout number with 2.0 mg/g WSAq-9 (**A**) but no significant increase in sleep bout length (**B**). Daily sleep time is not affected by WSAq-9 supplementation, with the exception of a reduction in nighttime sleep in males treated with the highest dose of 5.0 mg/g (**C**). WSAq-9treatment does not cause any changes in sleep pattern and time in females (**D**–**F**). Sleep was analyzed from age 42 d to 50 d. A one-way ANOVA l with Dunnett’s multiple comparisons test was used, and the number of analyzed flies is given in the bars. Bars represent means and error bars SEMs. * *p* < 0.05, ** *p* < 0.01 compared to controls on regular food.

**Figure 6 nutrients-14-03923-f006:**
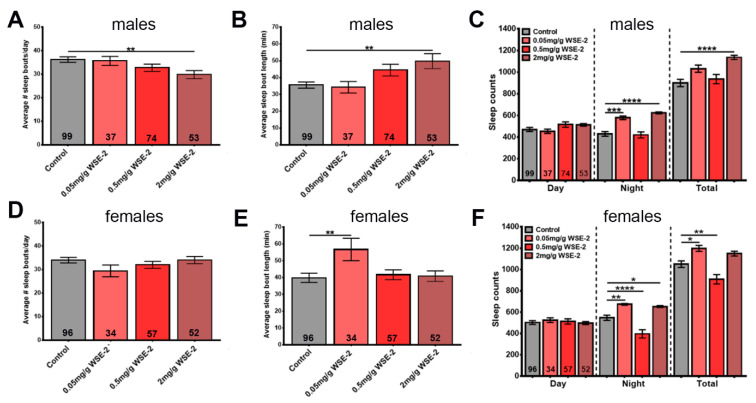
WSE-2 improves sleep in males. Males show a reduction in sleep bout number that reaches significance with 2.0 mg/g WSE-2 (**A**). This is accompanied by an increase in sleep bout length (**B**). WSE-2 treatment increases nighttime sleep and total daily time spent asleep (**C**). Females do not show a change in sleep bout number (**D**), and sleep bout length is only increased with the lowest concentration of 0.05 mg/g (**E**). The 0.05 mg/g and 2.0 mg/g doses increase nighttime sleep, and the lowest dose also increases total time spent asleep, whereas a reduction is seen in both with 0.5 mg/g (**F**). Sleep was analyzed from age 42 d to 50 d. A one-way ANOVA with Dunnett’s multiple comparisons test was used, and the number of analyzed flies is given in the bars. Bars represent means and error bars SEMs. * *p* < 0.05, ** *p* < 0.01, *** *p* < 0.001, **** *p* < 0.0001 compared to controls on regular food.

**Figure 7 nutrients-14-03923-f007:**
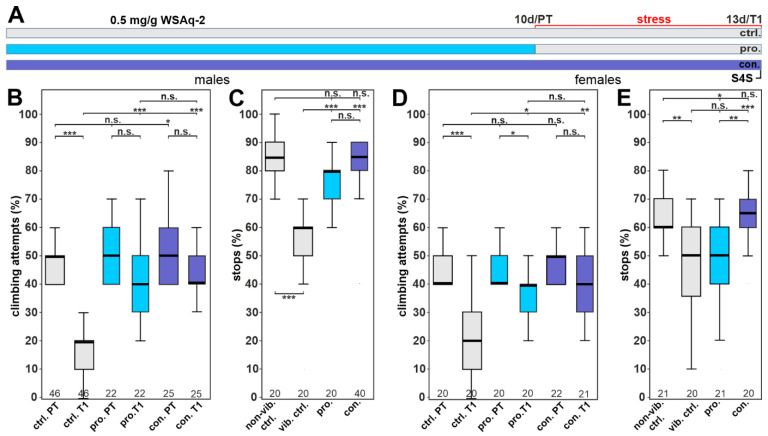
WSAq-2 protects against stress-induced behavioral phenotypes. (**A**) Schematic of the treatment paradigm (the time of WSAq-2 0.5 mg/g treatment) is indicated in blue/purple. (**B**) Percent of climbing attempts of males before (PT) and after (T1) stress was applied. (**C**) Number of stops males made at the sweet-tasting stripe. (**D**) Percent of climbing attempts of females before (PT) and after stress (T1) was applied. (**E**) Number of stops females made at the sweet-tasting stripe. A pairwise Wilcoxon test with Bonferroni–Holm correction for multiple comparisons was used in each panel. The number of analyzed flies is given below the boxes. The horizontal bars in the box plots represent the medians, boxes the 25% and 75% quartiles, and whiskers the data points within ± 1.5 times the interquartile range (IQR). Abbreviations: S4S = stop for sweet test, ctrl. = control, vib. ctrl = vibrated control, non-vib. ctrl. = non-vibrated control, pro. = prophylactic treatment, con. = continuous treatment. * *p* < 0.05, ** *p* < 0.01, *** *p* < 0.001, n.s. = not significant.

**Figure 8 nutrients-14-03923-f008:**
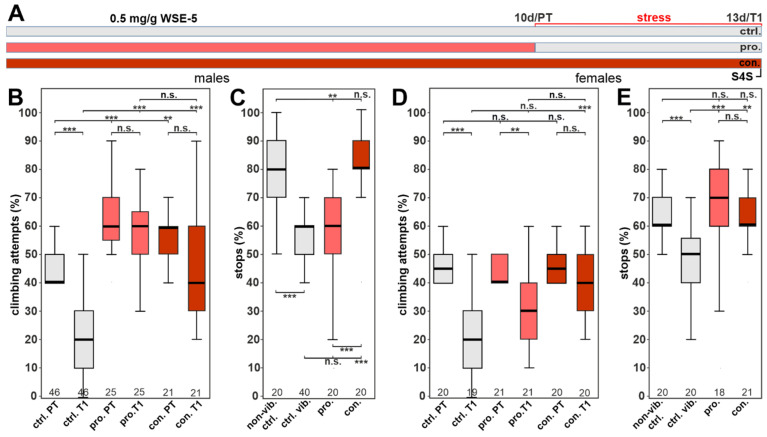
WSE-5 partially protects against stress-induced behavioral phenotypes at 0.5 mg/g. (**A**) Schematic of the treatment paradigm (the time of WSE-5) treatment is indicated in red. (**B**) Percent of climbing attempts of males before (PT) and after stress (T1) was applied. (**C**) Number of stops males made at the sweet-tasting stripe. (**D**) Percent of climbing attempts of females before and after stress was applied. (**E**) Number of stops females made at the sweet-tasting stripe. A pairwise Wilcoxon test with Bonferroni–Holm correction for multiple comparisons was used in each panel. The number of analyzed flies is given below the boxes. The horizontal bars in the box plots represent the medians, boxes the 25% and 75% quartiles, and whiskers the data points within ± 1.5 times the interquartile range (IQR). S4S = stop for sweet test, ctrl. = control, vib. ctrl = vibrated control, non-vib. ctrl. = non-vibrated control, pro. = prophylactic treatment, con. = continuous treatment. ** *p* < 0.01, *** *p* < 0.001, n.s. = not significant.

**Figure 9 nutrients-14-03923-f009:**
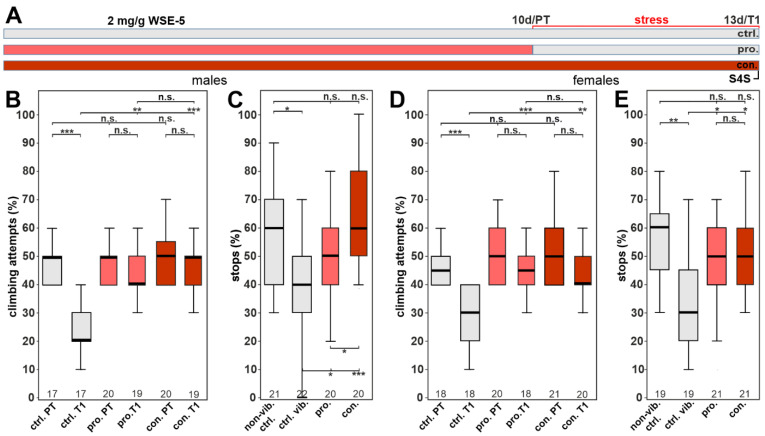
WSE-5 protects against stress-induced behavioral phenotypes at 2.0 mg/g. (**A**) Schematic of the treatment paradigm (the time of WSE-5 treatment is indicated in red). (**B**) Percent of climbing attempts of males before (PT) and after stress (T1) was applied. (**C**) Number of stops males made at the sweet-tasting stripe. (**D**) Percent of climbing attempts of females before and after stress was applied. (**E**) Number of stops females made at the sweet-tasting stripe. A pairwise Wilcoxon test with Bonferroni–Holm correction for multiple comparisons was used in each panel. The number of analyzed flies is given below the boxes. The horizontal bars in the box plots represent the medians, boxes the 25% and 75% quartiles, and whiskers the data points within ± 1.5 times the interquartile range (IQR). S4S = stop for sweet test, ctrl. = control, vib. ctrl = vibrated control, non-vib. ctrl. = nonvibrated control, pro. = prophylactic treatment, con. = continuous treatment. * *p* < 0.05, ** *p* < 0.01, *** *p* < 0.001, n.s. = not significant.

## Data Availability

Not applicable.

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
