# Peer review of "Withania somnifera Extracts Promote Resilience against Age-Related and Stress-Induced Behavioral Phenotypes in Drosophila melanogaster; a Possible Role of Other Compounds besides Withanolides"

_nutrients, 2022, doi:10.3390/nu14193923_

Round 1
Reviewer 1 Report
Traditional medicine has employed withania somnifera (WS) extracts to support healthy aging and wellbeing. In Western nations, WS is increasingly frequently taken as a nutritional supplement to promote resilience against age-related changes, such as sleep deprivation and depression. Although positive effects of WS have been supported by human studies, study designs have been very diverse. Plant material is inherently complex, and the origin of the plant material and the extraction technique have a significant impact on the extracts. Other components can be found in commercial supplements, and the research population's characteristics can also vary. The current study employed plant extracts that were composed and stablely examined in order to conduct studies that were as tightly regulated as possible. The study discovered that an ethanol extract of WS (WSE) improved sleep in old flies whereas a water extract of WS (WSAq) was most effective in enhancing physical fitness. Both extracts offered resistance to the behavioral alterations brought on by stress. Withanolides, which have been suggested as active components, were present in higher concentrations in WSE than in WSAq. Withanolides may therefore facilitate the improvement of sleep, whereas previously unidentified components that are enhanced in WSAq probably do so for the effects on fitness and behavior associated to stress. The current study is a well-controlled study and well-written. Just several minor concerns.
1, What’s the gender of the flies used in the current study, will gender affect the results?
2, What’s the potential mechanism contributing to the beneficial effects of withania somnifera (WS) either for improving sleep or enhancing physical fitness, which can be discussed a lit in the section of "Discussion".
Author Response
We thank the reviewer for his positive comment about our manuscript finding it " a well-controlled study and well-written." The comments are addressed below:
1, What’s the gender of the flies used in the current study, will gender affect the results?
We have analyzed males and females separately in each assay. The gender is indicated on top of each graph and effects on gender are described in the text. Briefly, beneficial effects of WS in the phototaxis assay are only significant in females and on sleep only in males. In the stress assays both, males and females are protected.
2, What’s the potential mechanism contributing to the beneficial effects of withania somnifera (WS) either for improving sleep or enhancing physical fitness, which can be discussed a lit in the section of "Discussion".
We have added a discussion of possible mechanisms that could play a role in the beneficial effects. However, determining if and which of these functions are playing a role in our assays would be beyond the scope of this manuscript.
Reviewer 2 Report
Activity data on crude extracts should have been thoroughly characterized by analysis of their major constituents (HPLC, NMR). This is a very major shortcoming of the paper.
Specific comments:
1. Scientific names such as "Drosophila melanogaster" should be italicized in the title.
2. In addition to ref [21-23], suggest authors to also cite a relevant and recent systematic review on the effects of WS supplementation on cognition (citation: pubmed.ncbi.nlm.nih.gov/31742775).
3. For the WS extracts, the method of extraction and the yield of dried extract as a percentage weight of the starting fresh or dried plant material should also be clearly stated.
4. Although the authors did deposit a voucher specimen, the value of the current report is greatly diminished by the lack of standardization and the fact that the authors did not characterize or analyse the plant extracts and their major constituents. This is a rather significant shortcoming.
5. Please state the level of significance used for the statistical analyses.
6. Please change "2-3 day" to "2 to 3 day", and try not to begin a sentence with a number.
7. "2-3 day old flies were collected and their wings shortened" - why was it necessary to shorten the wings?
8. Please state the ethical considerations of the present study and provide the actual institutional review board study/approval number. I find it hard to believe that "Institutional Review Board Statement: Not applicable".
9. The bioactive compounds in WS are thought to be mostly steroidal lactones. In particular, Withanolide A is a very hydrophobic molecule, practically insoluble (in water). This means that aqueous extracts would not contain Withanolide A. These important differences should be mentioned and correlated to the expected bioactive effects.
10. The discussion section should outline the important theoretical and mechanistic knowledge surrounding the potential clinical efficacy of WS. A clinical perspective is also lacking in the discussion section. Ashwagandha contains both water-soluble and fat-soluble bioactive components, as a result, ayurvedic practitioners may advise pairing it with milk. This should be at least briefly discussed.
11. Suggest having a separate conclusions section.
12. Authors should also suggest some areas for future research.
Author Response
- Scientific names such as "Drosophila melanogaster" should be italicized in the title.
This was an oversight and Drosophila melanogaster has now also been italicized in the title
2. In addition to ref [21-23], suggest authors to also cite a relevant and recent systematic review on the effects of WS supplementation on cognition (citation: pubmed.ncbi.nlm.nih.gov/31742775).
The reference has been added
3. For the WS extracts, the method of extraction and the yield of dried extract as a percentage weight of the starting fresh or dried plant material should also be clearly stated.
We have added information on the % yield of the dried extract.
4. Although the authors did deposit a voucher specimen, the value of the current report is greatly diminished by the lack of standardization and the fact that the authors did not characterize or analyse the plant extracts and their major constituents. This is a rather significant shortcoming.
We have added the analyses of the plant extracts as supplementary figure 1 and 2.
5. Please state the level of significance used for the statistical analyses.
The level of significance is indicated in each figure legend. It has now also been added to the methods.
6. Please change "2-3 day" to "2 to 3 day", and try not to begin a sentence with a number.
This has been changed
7. "2-3 day old flies were collected and their wings shortened" - why was it necessary to shorten the wings?
The behavioral assays depend on the flies walking. Therefore their wings are cut so that the cannot fly. This is now also mentioned in the methods.
8. Please state the ethical considerations of the present study and provide the actual institutional review board study/approval number. I find it hard to believe that "Institutional Review Board Statement: Not applicable".
This study only uses invertebrates (Drosophila) for which no ethical or animal care review is required. This is also in agreement with the requirement from NIH, which also does not require any institutional approval for working with Drosophila.
9. The bioactive compounds in WS are thought to be mostly steroidal lactones. In particular, Withanolide A is a very hydrophobic molecule, practically insoluble (in water). This means that aqueous extracts would not contain Withanolide A. These important differences should be mentioned and correlated to the expected bioactive effects.
We do find about significantly less Withanolide A in the water fraction with only 715ng/g of Withanolide A in the water fraction compared to 3448ng/g in the ethanol fraction (shown in the supplementary tables and in figure 1). The levels of other withanolides are also much lower (4-6 times less) in the water fraction (also shown in figure 1 and the supplementary tables). This is described in the text and this provides the rational for our hypothesis that Withanolides are not (or not the only) active compounds because the water extract id more protective in the fast phototaxis tests and also better in protecting from the stress-induced phenotypes. This is now discussed in more detail in the discussion.
10. The discussion section should outline the important theoretical and mechanistic knowledge surrounding the potential clinical efficacy of WS. A clinical perspective is also lacking in the discussion section. Ashwagandha contains both water-soluble and fat-soluble bioactive components, as a result, ayurvedic practitioners may advise pairing it with milk. This should be at least briefly discussed.
We are now briefly describing traditional use, clinical studies, and possible mechanistic functions in the discussion.
11. Suggest having a separate conclusions section.
We have added a short conclusion section
12. Authors should also suggest some areas for future research.
Possible future studies are described in the discussion
Reviewer 3 Report
- Title should be more precise. For example:
“Withania somnifera extracts promote resilience against age-related and stress-induced behavioral phenotypes in Drosophila melanogaster: the possible role of withanolides”
- Discussion
“In summary, our studies suggest that different compounds in WS may affect different behavioral outcomes, providing the need for future studies to identify these compounds.”
This point should more emphasized and discussed.
Author Response
- Title should be more precise. For example:
“Withania somnifera extracts promote resilience against age-related and stress-induced behavioral phenotypes in Drosophila melanogaster: the possible role of withanolides”
The title has been changed to
Withania somnifera extracts promote resilience against
age-related and stress-induced behavioral phenotypes in
Drosophila melanogaster; a possible role of other compounds be
sides withanolides
- Discussion
“In summary, our studies suggest that different compounds in WS may affect different behavioral outcomes, providing the need for future studies to identify these compounds.”
This point should more emphasized and discussed.
We agree that this is an important aspect of our manuscript and we have discussed this in more detail in the revised manuscript
Round 2
Reviewer 2 Report
Thank you for the revisions. Please note that scientific names such as "Drosophila" should be italicized throughout the manuscript.
Author Response
Please note that scientific names such as "Drosophila" should be italicized throughout the manuscript.
We have italicized the one "Drosophila" that was not already italicized (line482)